# A Lightweight Multi-domain Multi-attention Progressive Network for Single Image Deraining

Submission Id: 4803*

## ABSTRACT

Currently, the information processing in a spatial domain alone has intrinsic limitations that hinder the deep network's effectiveness (performance) improvement in a single image deraining. Moreover, the deraining networks' structures and learning processes are becoming increasingly intricate, leading to challenges in structural lightweight, and training and testing efficiency. We propose a lightweight multi-domain multi-attention progressive network ($M_2$PN) to handle these challenges. For performance improvement, the $M_2$PN backbone applies a simple progressive CNN-based structure consisting of the S same recursive $M_2$PN modules. This recursive backbone with a skip connection mechanism allows for better gradient flow and helps to effectively capture low-to-high-level/scales spatial features in progressive structure to improve contextual information acquisition. To further complement acquired spatial information for better deraining, we conduct spectral analysis on the frequency energy distribution of rain steaks, and theoretically present the relationship between the spectral bandwidths and the unique falling characteristics and special morphology of rain steaks. We present the frequency-channel attention (FcA) mechanism and the spatial-channel attention (ScA) mechanism to fuse frequency-channel features and spatial features better to distinguish and remove rain steaks. The simple recursive network structure and effective multi-domain multi-attention mechanism serve as the $M_2$PN to achieve superior performance and facilitate fast convergence during training. Furthermore, the $M_2$PN structure, with a small network component quantity, shallow network channels, and few convolutional kernels, requires only 168K parameters, which is 1 to 2 orders of magnitude lower than the existing SOTA networks. The experimental results demonstrate that even with such a few network parameters, $M_2$PN still achieves the best overall performance.

## CCS CONCEPTS

• **Computing methodologies → Image processing**; **Object detection**.

## KEYWORDS

Image Deraining, Lightweight, Multi-domain, Multi-attention.

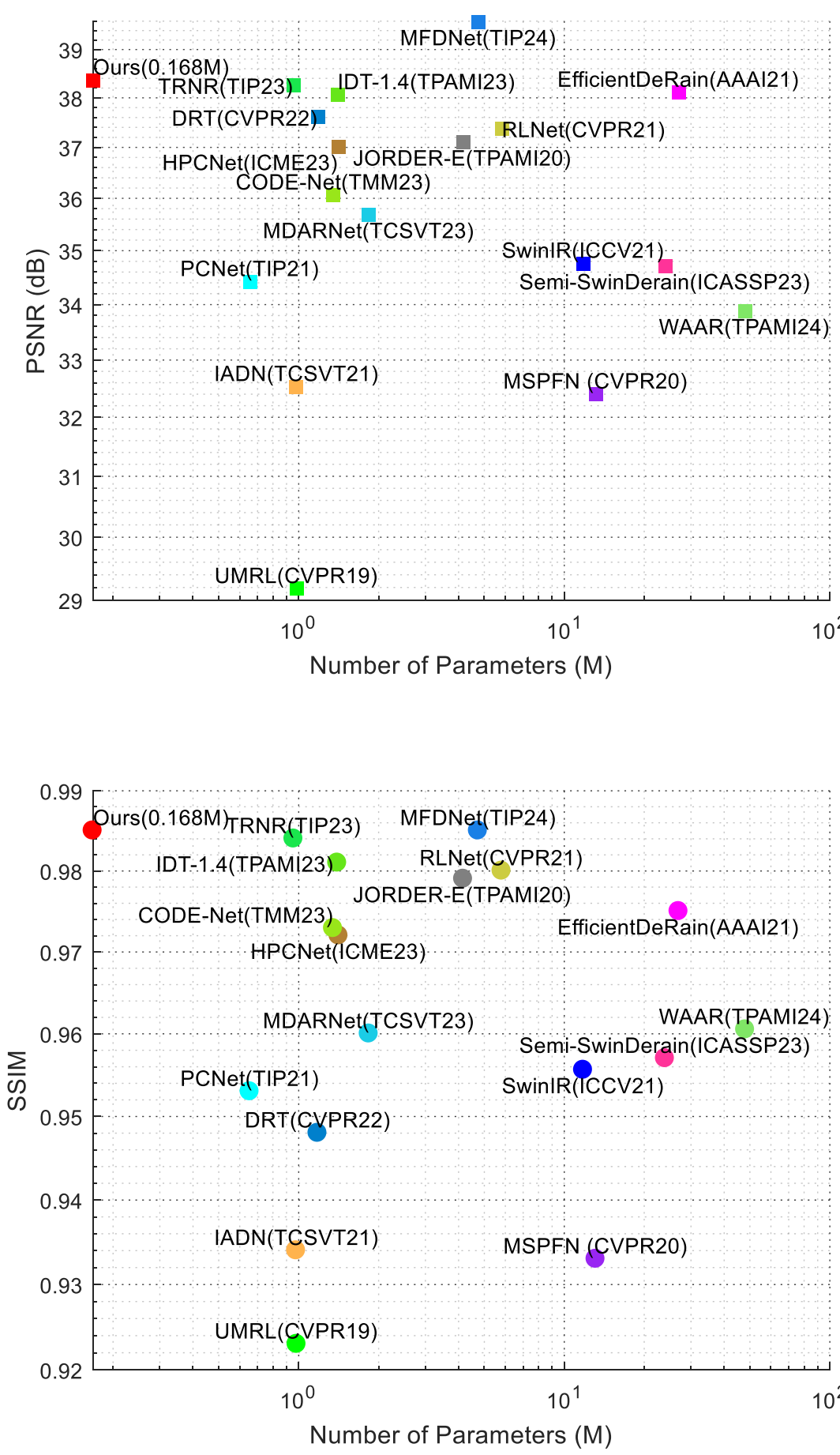

**Figure 1: The SOTA deep derainers' parameter quantity and performance comparisons on the Rain100L dataset.**

## 1 INTRODUCTION

Deraining algorithms aim to effectively remove the rain streaks or raindrops from the image, enhancing its visibility, and restoring its original quality. Currently, it is an active area of research to develop more robust and efficient deraining techniques, however, designing effective and efficient deraining algorithms remains a formidable challenge. Early deraining methodologies [1–4] for a single image heavily rely on statistical analyses of rain images, consequently rendering these hand-crafted priors constrained to address the varying and intricate rainy scenarios. In recent years, deep neural network (DNN) techniques, distinguished by their robust learning and generalization capabilities, have been introduced in the realm of deraining. The current deep-learning deraining (derainers), mainly including Adversarial learning (GAN) [5–9], CNN-based architectures [10–25], and Transformer architectures [26–35], have achieved remarkable improvements in performance. However, these deep derainers also sacrifice efficiency and discard lightweight architecture to achieve good performances.

Limitations of deep derainers: (i) Computational complexity and tremendous parameter issues. (i-a) Adversarial learning, such as DCD-GAN [8], typically requires training a generator and discriminator network in an alternating manner, which can be computationally demanding. (i-b) CNN-based derainers, such as RLNet

[16], require deep network depths (channels) and diverse Convolutional (*Conv*) kernels to extract hierarchical features and bolster abstraction capabilities, thereby causing a significant increase in parameters. (i-c) Transformer derainers with self-attention mechanisms, such as SwinIR [28], require calculating pairwise relationships between all pixels, which scales quadratically with the input image size, resulting in tremendous parameters. The tremendous network parameters exacerbate the challenge of parameter tuning and impeding convergence, and bring great limitations for real-time application and Internet-of-Thing (IOT) scenarios. The parameter quantities of state-of-the-art (SOTA) deep derainers are shown in **Fig. 1**. Moreover, the existed deep derainers still have some intrinsic limitations that prevent further deraining improvement.

Limitations of Adversarial derainers: (ii-a) difficult convergence, fine texture and spatial detail loss, side products of artifacts, weak generalization. Limitations of CNN-based derainers: (ii-b) limited receptive local field, weakly contextual information and global context modeling, fine texture and spatial detail loss. Limitations of Transformer derainers: (ii-c) limited local information processing capacity, fine texture, and spatial detail loss.

This work proposes a Lightweight Multi-domain Multi-attention Progressive Network ($M_2PN$) designed to strike an optimal balance between superior effectiveness (performance), high efficiency (short training period and testing time), and lightweight structure (few parameters). We describe the order of recursive $M_2PN$ structure—$M_2PN$ module—$M_2PN$ block as follows. (1) $M_2PN$ is a simple progressive CNN-based structure, consisting of the $S$ same recursive $M_2PN$ modules with skip connections, as shown in **Fig. 2**. The recursive structure with skip connection mechanism allows for better gradient flow and helps to capture low-to-high-level features effectively in progressive structure. When combined with recursion, this involves multiple levels of such connections within nested substructures of a progressive network, allowing for complex interactions across different representation levels/scales to improve contextual information (improve Limitation (ii-b/ii-c)). (2) Each $M_2PN$ module sequentially consists of $Conv_{(1\times1)}$ for shallow feature extraction, Long Short-Term Memory (LSTM) for an above skip recursion connection, an $M_2PN$ block-group through multiple domains and multiple attention for global field and local detail processing (improve Limitation (ii), and a $Conv_{(3\times3)}$ for feature fusion. (3) An $M_2PN$ block-group consists of the *same* $q$ $M_2PN$ blocks. Each $M_2PN$ block is a residual block with an original input-output branch and a parallel branch of frequency-channel-spatial feature attention and fusion. We have introduced the discrete cosine transform (DCT) principle to reveal the frequency energy distribution of the rain steaks, and then we have further analyzed the relationship between the spectral bandwidths of DCT and the unique falling characteristics and special morphology of rain steaks. This theoretical analysis contributed to proposing a novel frequency-channel attention (FcA) mechanism based on the characteristics of rain steaks. Furthermore, inspired by the energy function optimization principle presented in the literature [36], we introduced a fast closed-form energy solution without adding parameters to set up a 3-D spatial-channel attention (ScA) mechanism. The ScA mechanism learns the rainy information effectively in frequency-channel-spatial domains. Finally, only a negative SSIM Loss centrally guides the optimization process for a fast convergence.

Three main contributions are made: the local $M_2PN$, global progressive backbone, and well-designed structure, achieve good deraining effectiveness, high training and testing efficiency, and lightweight.

• This paper theoretically explains the relationship between the spectral bandwidths and rain steaks' unique falling characteristics and special morphology, and practically proposes a FcA mechanism. This FcA can decompose and recombine the distribution of different frequency energies in the rainy image into K-bandwidths and allows for learning rain steak's frequency information better to complement spatial information. Furthermore, the ScA mechanism provides a fast closed-form energy solution to fuse the acquired frequency-channel features in a spatial-channel domain for better identification and removal of rain steaks.

• The proposed $M_2PN$ progressive structure associates the contextual information through skip connections. Its residual $M_2PN$ block handles global information in the frequency domain, and the ScA mechanism focuses on local details. This holistic approach allows for focusing on the rain streak information from global to local scopes, spatial to frequency, and channel domains, that this progressive structure and fusion $M_2PN$ features can improve performance and also speed up network convergence (a short training period).

• The proposed $M_2PN$ has only 168K parameters owing to its shallow channels and few *Conv* kernels. $M_2PN$ backbone only includes $S\,(6) \times q\,(5) = 30$ $M_2PN$ blocks with only a maximum of 32 channels, and nearly parameter-free FcA and ScA. Owing to the expensive parameters of *Conv* computation, $M_2PN$ mainly uses the lightweight ShiftAddNet $Conv_{(3\times3)}$ [37] instead of traditional heavyweight $Conv_{(3\times3)}$, and 30 $Conv_{(1\times1)}$ for addressing fine detail.

## 2 RELATED WORK

### 2.1 Learning in Spatial and Channel Domains

The prior CNN-based derainers, for instance, JORDER [10] and the improved JORDER-E [11], are a recurrent deep learning architecture to joint detection and removal of rain streaks from the binary rain streak map, rain streak layers, and clean background. However, JORDER and its derivate JORDER-E trained on the binary rain streak map, may lead to loss of texture or sharp edges, especially when removing rain from images featuring heavy rain scenes. Subsequently, diverse CNN-based structures, which include multi-scale and multi-stream network [12], recursive multi-stage network without multi-scale [13], recurrent multi-scale networks [14, 15], recursive multi-scale (pyramid) networks [16–18], recursive multi-scale fusion architectures [19, 20], the encoder-decoder network [21, 22], have been proposed to pursue certain performance improvement. These CNN-based methods mostly adopt multi-scale structural schemes to extract and construct the hierarchical feature maps for coarse-to-fine deraining. To enhance the representation and capture crucial information, SPANet, IADN, and CODE-Net introduced spatial-attention [23] or channel-attention [24, 25] mechanisms to focus more on informative local regions or channels while suppressing less valuable ones. Recently, Transformer [26] and Swin-Transformer [27] with inherent multi-head attention mechanisms have been introduced to the deraining fields. SwinIR

[28], HPCNet [29], MFDNet [30], Semi-SwinDerain [31], DRT [32], IDT-1.4 [33], 4D-MGP-SRRNet [34], Restormer [35] redesigned and simplified the Transformer structure as the network backbones to enhance both local and global contextual information capturing capacity. However, Transformer derainers may not be as effective as convolutional layers in extracting and preserving the intricate local features that are crucial for accurately detailing rain streaks and maintaining sharp edges within an image.

## 2.2 Learning in Frequency-Domain

Image transforming from the spatial domain to the frequency domain for processing proves beneficial for conducting global spectrum analysis, efficient noise removal, compression, and filtering. Two related beneficial frequency analysis tools are wavelet and discrete cosine transformation (DCT). For example, RWL [38] integrated recurrent wavelet learning with visibility enhancement to perform scale-free deraining. SWAL [39] uniquely leverages the capability of wavelet transforms to separate an image's high-frequency and low-frequency information. WCAM [40] is a Wavelet Channel Attention Module that ingeniously substitutes the conventional down-sampling and up-sampling operations with wavelet transformation and its inverse. MDARNet [41] and WAAR [42] introduce hierarchy monogenic wavelet transform (HMW) and stationary wavelet (SW) transform to extract scale-space features and object edge features better, respectively. However, these HMW and SW transform components serve as independent preprocessing units, external to the MDARNet and WAAR network backbones, respectively. These non-end-to-end networks make parameter tuning and convergence challenging. Literature [43], based on the theoretical analysis of DWT (Haar) and DCT, integrates DWT into a network to propose Dual wavelet attention networks. Nevertheless, existing deep derainers integrated with DWT only decompose the image into LL, LH, HL, and HH subbands of level-2 to simplify feature extraction and information restoration (Inverse DWT). Considering the complex and diverse real-world rain scenes, the derainers employing just decomposition of level 2 (a fixed bandwidth) is far from sufficient to distinguish the energy of various frequency subbands in an image. Moreover, it is difficult to determine the cut-off frequency between the LL and HH of level 2. Literature [44] presents a learning-based (DCT) frequency selection method, which is another effective frequency analysis tool. Regrettably, few algorithms have fully utilized the powerful DCT tool to derive frequency domain analysis.

## 2.3 Learning in Lightweight

As shown in **Fig. 1**, the Transformer derainers generally have a large number of parameters, such as SwinIR with 11.8M [24], MFDNet with 4.741M [30], Semi-SwinDerain with 24M [31], 4D-MGP-SRRNet with 14.91M [31], Restormer with 25.31M [35]. Due to the limitations of the Q, K, and V structures and multi-headed self-attention mechanisms (Limitations (i-c)), derainers using the Transformer and its derivates as network backbones, are difficult to reduce the model parameters of the Transformer further. $\text{CNN}_{(s)}$ are also computationally intensive, and their models heavily rely on spatial Convolution to aggregate spatial information within images. The larger *Conv* kernels and deeper channels introduce a high computational load and lead to a sharp rise of parameters. Against this backdrop, a well-known method is Dilated Convolution [45], which uses a relatively small kernel size to get a large receptive field. However, there is no interdependence of the dilated *Conv* between different layers, which is fatal for pixel-level dense prediction. An alternative Depth-Wise Separable Convolution (*DWS-Conv*) [45] is aimed at reducing actual convolution channels by allowing each convolutional kernel to perform Convolution on just one channel. However, *DWS-Conv* lacks information exchange between different channels. Moreover, it spends more time than that computation on memory access, resulting in no reduction in actual computation even though the parameters are reduced. Recently, there are some novel convolution structures, e.g., AdderNet [46] with addition instead of multiplication, orthogonal ShiftNet with zero FLOPs and zero parameters [47], Deepshift [48] with shift and addition instead of multiplication, ShiftAddNet [37] with a fusion of shift operation and AdderNet for Edge device acceleration. Regrettably, these novel methods have not been applied in deep drainer to decrease computational costs.

# 3 LIGHTWEIGHT MULTI-DOMAIN MULTI-ATTENTION PROGRESSIVE NETWORK

This section illustrates a lightweight $\text{M}_2\text{PN}$ from a recursive $\text{M}_2\text{PN}$ structure—an $\text{M}_2\text{PN}$ block (from a global to local perspective).

## 3.1 Recursive $\text{M}_2\text{PN}$ Structure for Progressive Deraining

**Fig. 2** sketches the overall structure of the progressive $\text{M}_2\text{PN}$ consisting of the $S$ same recursive $\text{M}_2\text{PN}$ modules with skip connections, where $S$=6. The same recursive $\text{M}_2\text{PN}$ modules are used to gradually eliminate the rain streaks in the image at the $S$ stages. For simplification, the $1^{\text{st}}$ $\text{M}_2\text{PN}$ module has been taken as an example for description.

**First**, an RGB rainy image $x^{s-1}$ with a size of height $(H) \times$ width $(W) \times$ channle ($C$=3) and its copy are concatenated into the feature map $x_{in}^{s-1}$ with a size of $H \times W \times 6$ as the input of $\text{M}_2\text{PN}$, where $s$=1. Similarly, a copy of this rainy image $x^0$ in progressive $\text{M}_2\text{PN}$ structure, depicted with a green line in **Fig. 2**, is concatenated with the previous $\text{M}_2\text{PN}$ module $s$-1 output as input to the next $\text{M}_2\text{PN}$ module $s$, where $s$>1.

**Second**, at the initial stage of the $\text{M}_2\text{PN}$ module, a $Conv_{(1\times1)}$ as a preprocessing is used to extract shallow features from the input $x_{in}^{s-1}$, and obtained the $H \times W \times 32$ shallow feature map $f_{in}\ (x_{in}^{s-1})$, where $f_{in}\ (x_{in}^{s-1})$= $f_{in}\ (x^0 © x^{s-1})$=$Conv_{(1\times1)}\ (x^0 © x^{s-1})$.

**Third**, an LSTM [49] block in each $\text{M}_2\text{PN}$ module allows for complex interactions across different representation levels/scales to improve contextual information. LSTM also progressively passes on the processing results from the T $\text{M}_2\text{PN}$ modules, thereby continually refining and fine-tuning the intermediate deraining results at $S$ stages in frequency-channel-spatial domains. The flow of LSTM, including the equations of Input Gate $i_s$, Forget Gate $f_s$, Update Cell $u_s$, Output Gate $o_s$, Current Memory Cell $c_s$, and Hidden State $h_s$, are presented as Eq. (1).

$$\begin{cases} f_s = \sigma\left(W_f \cdot [x_s, h_{s-1}] + b_f\right), o_s = \sigma\left(W_o \cdot [x_s, h_{s-1}] + b_o\right) \\ u_s = \tanh\left(W_g \cdot [x_s, h_{s-1}] + b_g\right), c_s = i_s \odot u_s + f_s \odot c_{s-1} \\ h_s = \tanh\left(c_s\right) \odot o_s \end{cases} \quad (1)$$

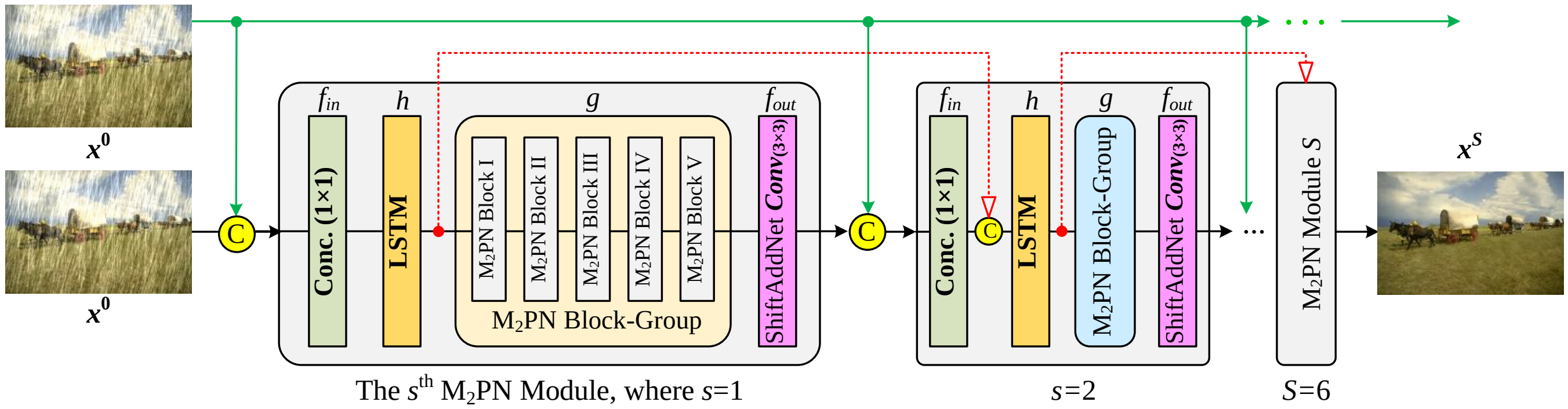

**Figure 2: Lightweight $M_2PN$ structure with $S$ recursion stages for progressive deraining.**

where $x_s = (f_{in}(x_{in}^{s-1}) \copyright h_{s-1})$ with the size of $H \times W \times 64$, is the input feature map of the LSTM layer, and the initially hidden state $h_{s-1}$ is set to a zero matrix with the size of $H \times W \times 32$.

**Fourth**, an $M_2PN$ block-group consists of the $q$ successive $M_2PN$ blocks. Each $M_2PN$ block, as described in subsection 3.2.2, can exploit a 3-D frequency-channel-spatial attention mechanism to extract and filter the fusion features in frequency-channel-spatial domains and output multi-domain and multi-attention feature map of $g(h(f_{in}(x^{s-1})))$.

**Fifth**, the enhanced deraining feature map $g$ is input into a ShiftAddNet $Conv_{(3\times3)}$ layer for feature reconstruction, the rain removal feature is restored to the RGB color space image, and the deraining results of the current iteration stage are output $x^s$.

As shown in **Fig 2**-(a), the correlation between the input and output of the $M_2PN$ module at stage $s$ can be expressed in equation (2), where $f_{out}$ function is a ShiftAddNet $Conv_{(3\times3)}$ operation.

$$x^s = x^0 + f_{out}\left(g\left(h\left(f_{in}(x^{s-1})\right)\right)\right) \tag{2}$$

## 3.2 Multi-domain Multi-attention ($M_2PN$) block for rain feature extraction and processing

### 3.2.1 *The benefits of DCT characteristics on deraining.*

An image's content is represented by its distribution and relative variations of its pixels. In the spatial domain, the textures or edges of rain steaks in the image reflect the rapid change of pixel value of the local region, which can be represented by using the magnitude of the gradient. In the frequency domain, a spatial image can be transformed into a spectral map, indicating the image's DC, low-medium-high frequency energy distributions. Particular emphasis is placed on the textures of rain steaks, which are associated with high-frequency energy.

Regrettably, most existing deraining schemes still overlook or do not pay attention to frequency analysis technologies. Moreover, the only existing individual research schemes that try to use frequency analysis technology have two limitations. (1) Neither wavelet using mean operation [39–43] nor high-pass filters [50] lacks effective local analysis capacity. (2) Due to the rich objects and complex background information, it is difficult to determine a single cut-off frequency in either wavelet or high-pass filters to segment the low-frequency and high-frequency bandwidths effectively. (3) Those wavelet schemes mainly perform spectrum decomposition in vertical and horizontal directions, neglecting rain steaks' morphology and drop characteristics.

As everyone knows, DCT can span the spectrum from DC to high-frequency bandwidths, thereby introducing the DCT principle and integrating the characteristics of rain steak, which are expected to overcome three limitations.

**(1) DCT with both local and global spectral analysis capacity**

The expression of DCT is given in Eq (3).

$$Freq^n = D\sum_{i=0}^{N-1}\sum_{j=0}^{N-1} g_{i,j}^{2d}\cos\left(\frac{\pi u}{N}\left(i+\frac{1}{2}\right)\right)\cdot\cos\left(\frac{\pi v}{N}\left(j+\frac{1}{2}\right)\right),$$
$$where \quad D = \begin{cases}\sqrt{\frac{1}{N}}, & u = 0\ or\ v = 0\\ \sqrt{\frac{2}{N}}, & u \neq 0\ or\ v \neq 0\end{cases} \tag{3}$$

where the parameter $N$ is the height or width of the segmentation square block, the $i$ and $j$ represent the location of the $i^{\text{th}}$ row and the $j^{\text{th}}$ column in the segmentation block, $g_{i,j}^{2d}$ is the pixel or feature value in the segmentation block. The most important parameters, $u$ and $v$, are the vertical and horizontal spectral decomposition components, and D is a constant associated with $u$ and $v$.

As Eq. (3) implies, the DCT steps may include segmenting the image into blocks, applying DCT to transform spatial information into spectrum/frequency feature in each block, which reflects local spectral analysis capacity, and then analyzing and removing the frequency components related to rain streaks in the frequency domain. A DCT spectrum matrix covers all the frequency domain statistics of a rainy image, and that DCT matrix reflects robust global spectral analysis capacity.

**(2) DCT with spectral decomposition and transformation capacity from multi-direction**

As Eq. (3) implies, DCT can mine spatial image information and decompose into a series of spectral bandwidths by varying the parameters $u$ and $v$. Furthermore, DCT can transform and decompose information from multi-directions, overcoming the defect of a single cut-off frequency in both wavelet and high-pass filters. However, this raises a new question: which combinations of $u$ and $v$, or which spectral bandwidths, are more suitable for deraining?

**(3) Relationship between DCT's spectral bandwidths and the rain steaks' unique falling characteristics and special morphology**

To determine the parameters $u$ and $v$, DCT is applied to the decomposition of both the rain-free image and the corresponding rainy image, observing the variations in their frequency feature maps.

As shown in **Fig. 3**, the $4^{th}$ and $5^{th}$ columns are the DCT frequency maps, namely, spectral feature maps. Two points, namely, spectral bandwidth and frequency energy variation in the spectral feature maps, require consideration. Spectral bandwidth: the top left corner of the spectral feature map represents the DC component, and the other parts are the AC component. The higher vertical and horizontal frequency components or spectral bandwidths are distributed along the $u$-axis and $v$-axis. The corners of the bottom left, top right, and bottom right represent the highest vertical, horizontal, and combined spectral bandwidths. Frequency energy variation: the red color labeled 10 in the spectral feature map represents the most active energy, and vice versa.

Known that, influenced by the Earth's gravity, rain streaks generally fall vertically or in an approximate vertical direction, thereby, rain streaks in images typically manifest a certain degree of directionality and repetitiveness. **Fig. 3** depicts three common phenomena of rain streak patterns. The $1^{st}$ and $2^{nd}$ columns are the rain-free images and the corresponding rainy image, and the $3^{rd}$ column depicts three common simulated falling and morphology of rain steaks.

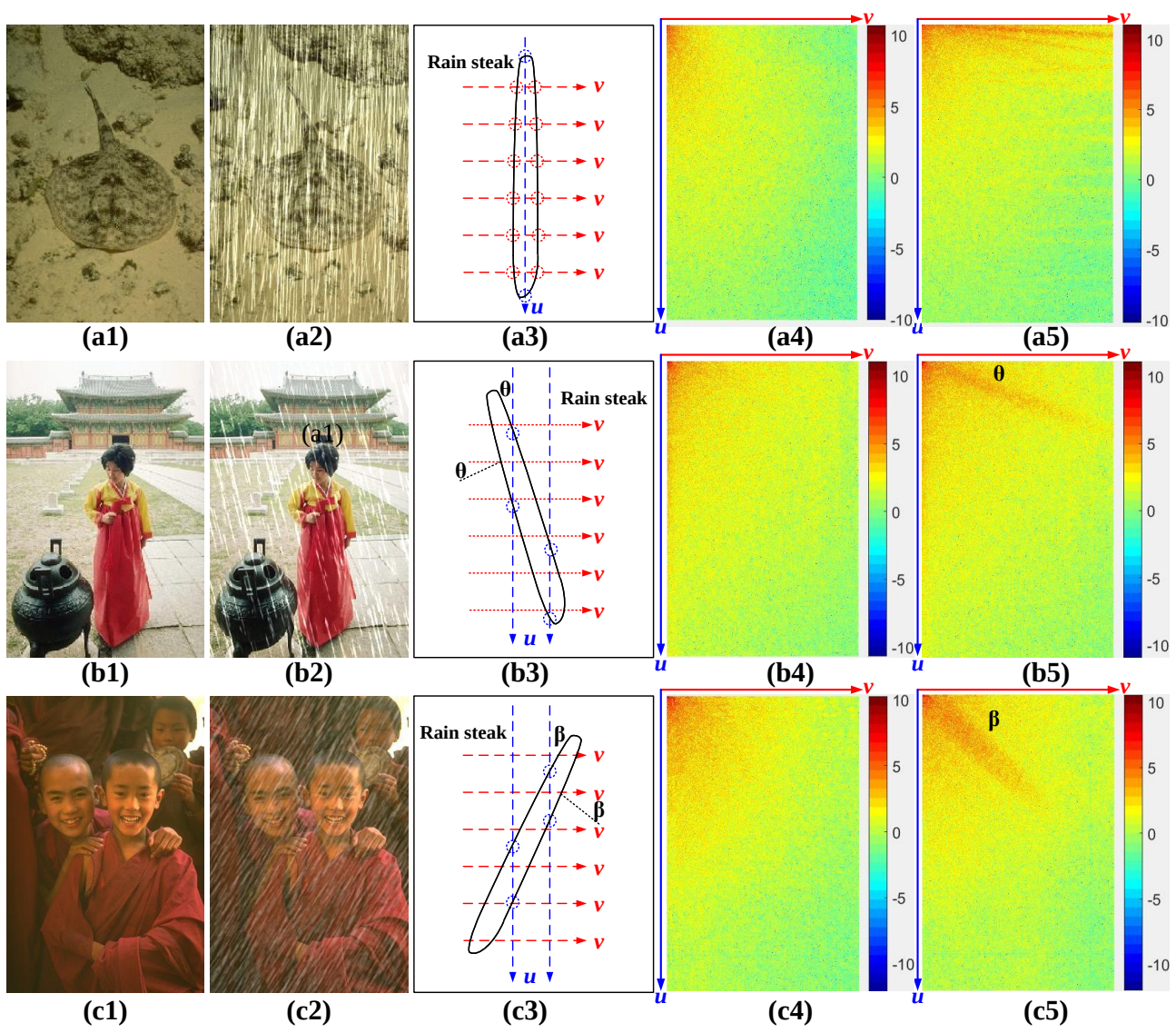

**Figure 3: The $1^{st}$ and $2^{nd}$ columns are the rain-free images and the corresponding rainy images, and the $4^{th}$ and $5^{th}$ columns are the corresponding DCT feature maps.**

The first phenomenon is the approximately vertical falling rain steak, as shown in **Fig. 3**–(a2). **Fig. 3**-(a3) shows the simulated falling and morphology. **Fig.3**-(a4) shows a DCT map of a rain-free image that follows the basic DCT characteristics, where the top left corner contains mostly DC and low-frequency components, concentrating most of the image energy (shown in approximate red color). As one moves toward the bottom right corner, the frequency increases while the energy decreases accordingly. Compared to the DCT map of the rain-free image, the active energy in **Fig. 3**-(a5), represented in red color, manifests along the horizontal $v$-axis, progressing from low to high frequencies. The reason for the energy variation is simple: First, the textures of steaks have high-frequency bandwidth, and the relatively smoothed interior regions mainly refer to low-frequency energy. It means that leveraging the properties of the DCT to address local rain streak texture involves transitioning from low-high-low-frequency transformation. Second, the rain steak with the elongated shape in the vertical direction means that the frequency decomposition mainly goes along the horizontal $v$-axis, thereby, it is clear that the spectral bandwidth and energy variation of vertically falling rain steak mainly varies in the horizontal $v$-direction of DCT.

The second phenomenon is at a $\theta$ angle to the vertical falling rain steak, as shown in **Fig. 3**–(b2), where $\theta$ is smaller than $45°$. This is a common rainy phenomenon where the simulated falling and morphology are shown in **Fig. 3**-(b3). The frequency component and energy distributions of this rain-free image in **Fig. 3**-(b3) are similar to **Fig. 3**-(a4). It is observed from **Fig. 3**-(b5) that the energy variation is at an approximately $\theta$ angle to the horizontal $v$-axis, progressing from low to high frequencies. It is clear that the falling direction $\theta$ and special morphology of rain steak determine the energy variation direction $\theta$ and spectral bandwidth in DCT. The third phenomenon is similar to the second one in that the rain steak is at a $\beta$ angle to the vertical falling direction, as shown in **Fig. 3**–(c2) and (c3), where $\beta$ is also smaller than $45°$. Other falling directions of rain steak, such as horizontal falling or largely deviating from vertical falling, rarely occur in natural phenomena.

In summary, the approximate vertical falling or slightly deviating from the vertical falling and elongated morphology of rain steaks can determine the combination of $u$ and $v$ in DCT, which concentrates in the upper right portion of the DCT frequency matrix (UR-DCTM). Through appropriate UR-DCTM frequency filtering, rain streaks can be separated from other contents in the image, thereby achieving rain removal.

**(4) Determination of UR-DCTM coefficients** Based on the analysis mentioned above, the preliminary range of UR-DCTM coefficients is enclosed by the green polygon formed by 28 boxes in **Fig. 4**-(c). However, considering the falling characteristics, the phenomenon of raindrops falling at a $45°$ tilt is rarely observed. The range of UR-DCTM coefficients can be further narrowed down to the blue polygon formed by 22 boxes. To further narrow down the coefficient range, the usage of DCT spectral bandwidth for frequency filtering in vision tasks requires analysis. Recently, the literature [44] provided a meaningful reference to determine the effective combination of $u$ and $v$. **Fig. 4**-(a) and (b) show the DCT frequency heat maps of Y (Luminance) on ImageNet and COCO datasets, respectively. As shown in **Fig. 4**-(a) and (b), the DC and low-frequency bandwidth (boxes with small indices) carry more activated energy and informative contents (colors close to dark red), than the high-frequency bandwidth (boxes with large indices). This observation highlights that, in general, the selection of the low-frequency bandwidth is generally much more suitable for applying for vision inference tasks than those of high-frequency. Combining observation of **Fig. 3**-(a4) (c4) and (a4) (c5), the highest frequency energy located at the $v$-axis and moving along the $u$-axis is relatively inactivated compared to the low-frequency. These relatively

inactivate energies map into the right edge of **Fig. 4**-(c), namely, the boxes with (27, 28, 38, 39, 45, 46) indices. After removing these six coefficients, the rest sixteen are UR-DCTM coefficients, which are indices with (0, 0) and (0, 1) indicated in dark red and (5, 6, 7, 13, 14, 15, 16, 17, 25, 26, 29, 30, 37, 40) in red.

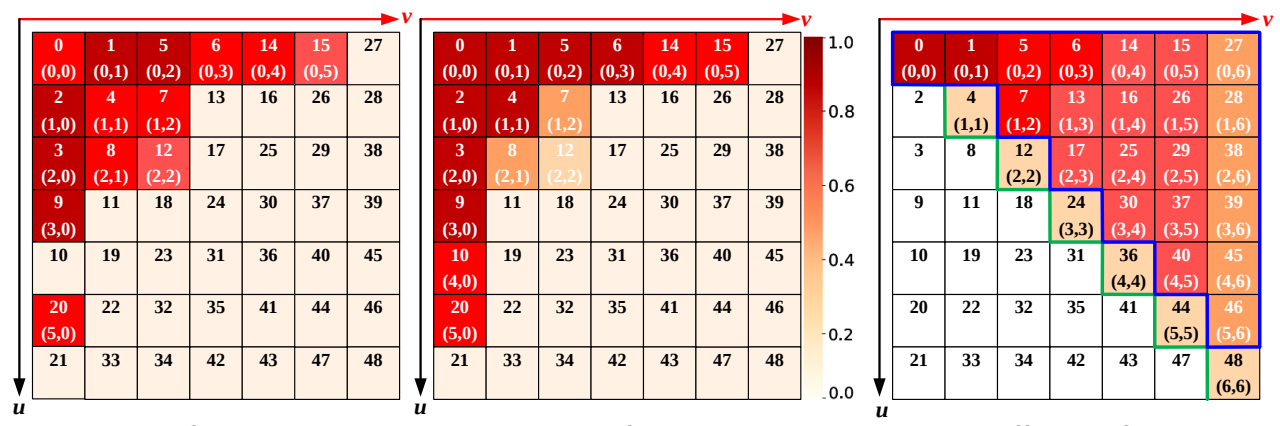

(a) Heat maps of Y on COCO dataset (b) Heat maps Of Yon the ImageNet dataset (c) Coefficients of UR-DCTM

**Figure 4: Heat maps of Y on COCO, ImageNet dataset, and heat maps to determine UR-DCTM coefficients.**

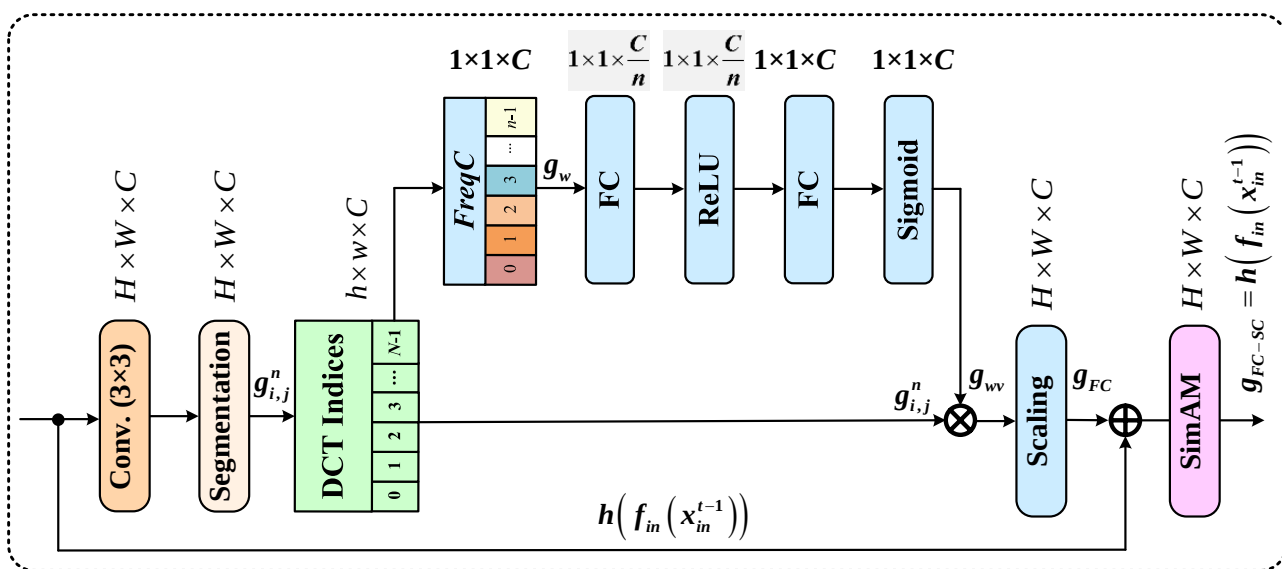

**Figure 5: Multi-domain Multi-attention ($M_2$PN) block for rain feature extraction and processing.**

#### 3.2.2 *The $M_2$PN block with FcA and ScA mechanism for rain feature extraction and processing.*

Each $M_2$PN block is a residual structure combining frequency, channel, and spatial attention. A $M_2$PN block includes five stages, showing with five colors in **Fig. 5**.

(1) Convolution: A $Conv_{(1\times1)}$ is used to extract the features in a feature map $g(f_{in}(x_{in}^{t-1}))$ with a size of $H \times W \times 32$, which enhances and enriches the rain texture feature extraction.

(2) Segmentation: an adaptive global average pooling (GAP) is applied to reduce the spatial size of these feature sub-models from $(H \times W \times 32)$ to $(h \times w \times 32)$. Then, the 32 channels are segmented into $n$ equal parts to obtain $n$ sub-models $g_{i,j}^{n}$ $=g^0, g^1, ..., g^{n-1}$.

(3) DCT indices and matrix: DCT with the combination of $u$ and $v$ in Eq. (1) is applied to obtain multiple frequency transformation or decomposition coefficients to set up a DCT frequency matrix (DCTM).

(4) Frequency-channel attention (FcA): The $n$ coefficients in UR-DCTM, as shown in **Fig. 4**-(c), are concatenated into a $n$-frequency-channel attention vector $FrecC = Cat(FreqC^0, FreqC^1, \dots, FreqC^{n-1})$, where $n$ is 16. As mentioned in subsection 3.2.1, the well-designed *FreqC* vector is based on considering both unique falling characteristics and the special morphology of rain steaks, and its spectral bandwidths contain the DC-low-high frequency. The *FreqC* vector is element-wise multiplied with the $n$ sub-models $g_{i,j}^{n}$ =to obtain a weighted deraining feature map $g_w = Freq \odot g_{i,j}^{n}$, where $i$ and $j$ represent the element of the $i^{\text{th}}$ row and $j^{\text{th}}$ column in each sub-model. Then, a fully connected layer (FC) and a Rectified Linear Unit (ReLU) are used to transform the weighted deraining feature map $g_w$. This transformation process involves summing the spatial dimensions of the feature subgraph, resulting in a spatially dimension-normalized feature map with dimensions of $1 \times 1 \times C$. In the FC layer, the $C$ channels are evenly divided into $n$ parts, transforming the $C$ channels into $C/n$ channels, thereby reducing computational load. These channels are then activated through the following activation ReLU function. Subsequently, another FC layer converts the $C/n$ partitioned channels back to the original $C$ channels, thus obtaining the transformed feature sub-map. A Sigmoid layer activates the sub-map to get the weight vector $g_{wv}$ of frequency-channel attention, where $g_{wv}$ = Sigmoid(FC(ReLU(FC($g_w$))). The weight vector $g_{wv}$ with the size of $1 \times 1 \times C$ is weighted onto the features of each channel of sub-models $g_{i,j}^{n}$, subjected to a scaling (scale) operation, and then the enhanced frequency-channel attention (FcA) map $g^{FC}$ is outputted.

(5) Spatial-channel attention (ScA): The attention mechanism is important to spatial information, owing to the suppression of spatial redundant information and the increase in efficiency. However, the common 1-D channel attention and 2-D spatial attention mechanisms are relatively independent and inefficient. Yang *et al.* [36] have built upon renowned neuroscience theories, derived a rapid closed-form solution for the energy function, and finally proposed a simple, parameter-free SimAM module.

$$e_s = \frac{4(\sigma^2 + \lambda)}{(t-\mu)^2 + 2\sigma^2 + 2\lambda} \quad (4)$$

where SimAM is a ScA mechanism, $s$ is the target neuron, $\mu$ and $\sigma$ are the mean and variance of the neurons surrounding $s$ in a channel, $\lambda = 0.0005$ is a constant, and $e_s$ is a target neuron energy. The smaller $e_s$ is, the more distinction from surround neurons is. Eq. (5) presents a 3-D $(H \times W) \times C$ attention fusing the spatial and channel domains. Then, all neurons form into group $E$, which connects spatial and channel dimensions. Finally, a Sigmoid function activates energy across three dimensions and restricts values that are too large in group $E$.

$$g_{FC-SC} = \text{Sigmoid}(1/E) \odot g_{FC} \quad (5)$$

where $g_{FC-SC}$ is the final output of an $M_2$PN block, the correlation between the input and output of an $M_2$PN block is

$$g_{FC-SC} = \text{ScA}\left(h\left(f_{\text{in}}\left(x^{s-1}\right)\right) + g_{FC}\right) = g\left(h\left(f_{\text{in}}\left(x^{s-1}\right)\right)\right)$$

The benefit is that the added ScA without the extra parameters can further learn the important information from the spatial-channel domains to complement frequency-channel information through a well-designed cross-modality fusion block.

## 3.3 Loss Function

To further reduce the parameters and speed up convergence, a simple negative Structural Similarity Index (SSIM) loss [11, 33] facilitates the training of the progressive $M_2$PN. For an output image of the $M_2$PN structure with $S$ progressive stages, there are $S$ output image $x^s$ and the ground-truth $x_{GT}$. The negative SSIM

function is listed in Eq. (6).

$$L = -\sum_{s=1}^{S=6} \omega_s \, \mathrm{SSIM}\left(x^s, x_{GT}\right) \tag{6}$$

where $\omega_s$ is a balance parameter for recursive supervision.

## 4 EXPERIMENTS

### 4.1 Experimental Settings

**Evaluation Datasets and Metrics.** Five benchmark datasets are Rain100L, Rain100H, Rain200L, Rain200H [10], and Rain1400 (DDN-Data) [50], for the $M_2$PN training and experimental evaluation. Rain100L, Rain100H, Rain200L, Rain200H, Rain800, and Rain1400 datasets provide 200, 1800, 1800, 1800, and 12600 images with a total of 18200 training images. They also provide 100, 100, 200, 200, and 1400 images with a total of 2000 testing images. Peak Signal-to-Noise Ratio (PSNR) and SSIM are effective evaluation metrics. The higher the PSNR and SSIM scores are, the better the performances are.

**Comparison Methods**. This paper presents seventeen SOTA methods for comparisons, including JORDER-E [11], UMRL [14], TRNR [15], RLNet [16], MSPFN [19], PCNet [20], EfficientDeRain [22], IADN [24], CODE-Net [25], SwinIR [28], HPCNet [29], MFD-Net [30], Semi-SwinDerain [31], DRT [32], IDT-1.4 [33], MDARNet [41], WAAR [42].

**Implementation Details.** The initial learning rate ($lr$) is $4 \times 10^{-4}$ and decreased with a factor of 0.2 at 30, 50, and 80 epochs, and the total epochs of 100 is a much shorter training period. The optimizer Adam, with a batch-size of 18, is used to train the $M_2$PN model on a single NVIDIA 3060-12G GPU.

**Table 1: The ablation studies of $M_2$PN and its three variants.**

| Derivatives | *w/o* LSTM | *w/o* DCT | *w/o* SimAM | $M_2$PN |
|---|---|---|---|---|
| PSNR↑ | 36.95 | 35.28 | 37.92 | 38.01 |
| SSIM↑ | 0.971 | 0.965 | 0.985 | 0.985 |
| Para.(K)↓ | 93.8K | 167.5K | 167.6K | 167.6K |

### 4.2 Ablation Study

A series of ablation studies are conducted on the structural block of the $M_2$PN to demonstrate the effectiveness of the main components, which mainly include the skip connection in the LSTM layer, the frequency-channel attention processing in the DCT layer, 3-D spatial-channel attention processing in the SimAM layer, are presented in discussions. The trained and tested ablation studies are performed on the Rain100L dataset. Three $M_2$PN variants, by removing those individual components, in turn, are denoted as *w/o* LSTM, *w/o* DCT layer, and *w/o* SimAM. Table 1 shows the comparisons between the effectiveness of the main components of $M_2$PN. First, it is observed that $M_2$PN achieves the best deraining performance compared to other ablation variants, with only generating slight parameters additionally. Second, the additional DCT and SimAM layers are almost parameter-free components that contribute to the designed lightweight network. The $M_2$PN with the DCT layer and SimAM layer realize significant performance improvements, where the DCT layer with only 128 parameters raised at 2.7 dB and 0.02 (2%) in terms of PSNR and SSIM compared to the *w/o* DCT layer. The SimAM layer achieves a similar performance improvement without additional parameters. Third, the LSTM layer allows for complex interactions across different representation levels/scales to improve contextual information and finally contributes to the progressive rain steak removal.

### 4.3 Evaluations with SOTA methods across multiple datasets

The experimental evaluations are conducted on seventeen SOTA methods across five synthetic benchmark datasets, as listed in Table 2. For a comprehensive evaluation of $M_2$PN and SOTA methods in the deraining field, the experimental results are conducted in three aspects: deraining effectiveness (performance), efficiency (training periods), and lightweight (network parameters).

**Effectiveness Comparisons.** Table 2 indicates the top-three (PSNR or SSIM) scores in each column in bold letters with red, green, and blue colors. There are two metrics, PSNR and SSIM, for performance evaluation on every dataset, thereby, five datasets have ten evaluation items. First, our $M_2$PN, TRNR [15], MFDNet [30], MSPFN [19], and EfficientDeRain [22] get 7, 7, 7, 4, and 3 items in the top-three scores. Second, to precisely evaluate the performances, top-three (PSNR or SSIM) scores indicated with red, green, and blue colors are labeled 3, 2, and 1 scores respective to the 1st, 2nd, and 3rd places. The TRNR [15], our $M_2$PN, MFDNet [30], EfficientDeRain [22], and MSPFN [19] get 16, 13, 13, 8, and 8 performance scores. From the statistical performance results, it can be seen that our $M_2$PN is one of the best deraining models. **Fig. 6** shows the deraining quality comparison between these top four models. The deraining effects of these top four models are good and similar, and it is difficult to distinguish the differences between them through the naked eye.

**Efficiency Comparisons.** The PCNet [16], our $M_2$PN, and MDARNet [41] only require 60, 100, and 120 training epochs, respectively. This means that our $M_2$PN only requires 100 epochs to make the network convergence and ensures a good fitting between the training and testing performances. While the PCNet [16] and MDARNet [41] have comparable training epochs to our $M_2$PN, our $M_2$PN distinctly outperforms theirs in deraining effectiveness. Attributed to the multi-attention mechanism concerning important multi-domain information, our $M_2$PN can achieve excellent deraining performance even with a shorter training period. Furthermore, our $M_2$PN takes only 0.11 seconds to remove rain streaks from a single rainy image with a size of $512 \times 512$. The testing time of $M_2$PN is shorter than SwinIR of 0.81s, EfficientDeRain of 0.4s, TRNR of 0.5s, and MSPFN of 0.59s.

**Lightweight Comparisons.** Our $M_2$PN contains the minimal parameters of only 168K over other deraining networks, followed by PCNet with 0.656K and TRNR with 0.958K, respectively. The small number of network parameters simplifies parameter tuning and speed-up convergence. It will contribute greatly to real-time applications and IOT scenarios.

**Summary.** Our $M_2$PN model achieves the 2nd best deraining performance with the 2nd shortest training period and the most lightweight structure. Taking into account the comprehensive factors of effectiveness, efficiency and lightweight attributes, the comprehensive ranking is our $M_2$PN, TRNR [15], MFDNet [30], EfficientDeRain [22], and MSPFN [19], which get the 18, 17, 13, 8, 8 scores, respectively.

**Table 2: The comprehensive comparisons with SOTA methods**

| Datasets | Rain100L | | Rain100H | | Rain200L | | Rain200H | | Rain1400 | | Para. | Epoch↓ | Score↑ |
|---|---|---|---|---|---|---|---|---|---|---|---|---|---|
| Metrics | PSNR↑ | SSIM↑ | PSNR↑ | SSIM↑ | PSNR↑ | SSIM↑ | PSNR↑ | SSIM↑ | PSNR↑ | SSIM↑ | (M)↓ | | |
| JORDER-E | 37.10 | 0.979 | 24.54 | 0.802 | 36.90 | 0.973 | 28.58 | 0.876 | 32.08 | 0.911 | 4.17 | 135 | 0 |
| UMRL | 29.18 | 0.923 | 26.01 | 0.832 | — | — | — | — | — | — | 0.984 | — | 0 |
| TRNR | **38.26** | **0.984** | 30.09 | **0.910** | **39.50** | **0.986** | **29.85** | **0.918** | 30.66 | 0.917 | **0.958** | 888 | **17** |
| RLNet | 37.38 | 0.980 | 28.55 | 0.860 | 36.64 | 0.971 | 28.87 | 0.895 | 30.72 | 0917 | 5.82 | 240 | 0 |
| MSPFN | 32.40 | 0.933 | 28.66 | 0.860 | **39.48** | **0.984** | 29.66 | 0.890 | **32.93** | **0.923** | 13.35 | 500 | **8** |
| PCNet | 34.42 | 0.953 | 28.45 | 0.870 | — | — | — | — | — | — | **0.655** | **60** | 5 |
| EfficientDeRain | 38.11 | 0.975 | **30.88** | **0.905** | 38.20 | 0.978 | **30.58** | 0.900 | 32.15 | 0.922 | 27 | 10000 | **8** |
| IADN | 32.53 | 0.934 | 27.86 | 0.835 | 33.12 | 0.940 | 28.22 | 0.845 | 30.05 | 0.878 | 0.98 | **100** | 2 |
| CODE-Net | 36.06 | 0.973 | 27.50 | 0.849 | 36.98 | 0.976 | 27.93 | 0.863 | 31.56 | 0.901 | 1.35 | 258 | 0 |
| SwinIR | 34.76 | 0.956 | 29.09 | 0.872 | 37.61 | 0.975 | **29.70** | 0.885 | 31.16 | 0.911 | 11.8 | 1200 | 1 |
| HPCNet | 37.01 | 0.972 | **30.12** | 0.893 | — | — | — | — | — | — | 1.416 | 500 | 2 |
| MFDNet | **39.58** | **0.985** | **30.48** | **0.899** | **38.87** | 0.982 | 29.61 | **0.908** | **32.89** | 0.922 | 4.741 | 600 | **13** |
| Semi-SwinDerain | 34.71 | 0.957 | 28.79 | 0.861 | — | — | — | — | 32.68 | **0.932** | 24 | 5000 | 3 |
| DRT | 37.61 | 0.948 | 29.47 | 0.846 | 38.30 | 0.928 | 28.91 | 0.859 | 32.86 | 0.920 | 1.18 | 4600 | 0 |
| IDT-1.4 | 38.06 | 0.981 | — | — | 38.52 | **0.983** | 28.91 | 0.893 | — | — | 1.4 | — | 1 |
| MDARNet | 35.68 | 0.960 | 29.71 | 0.884 | — | — | — | — | — | — | 1.84 | 120 | 0 |
| WAAR | 33.88 | 0.961 | 24.78 | 0.823 | — | — | — | — | — | — | 48.1 | — | 0 |
| $M_2PN$ | **38.36** | **0.985** | 29.36 | **0.899** | 37.90 | **0.983** | 28.81 | **0.900** | **32.95** | **0.923** | **0.168** | **100** | **18** |

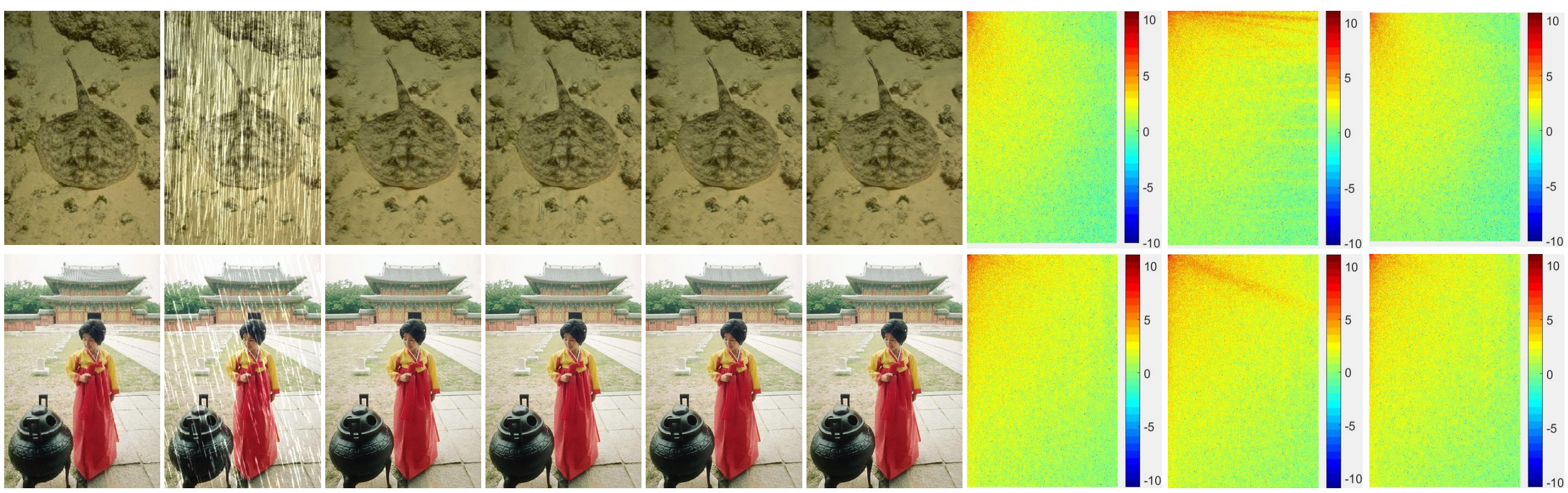

Grongd-truth (GT) Rainy Image EfficientDeRain MFDNet TRNR $M_2PN$ GT Spectrum Rainy Image Spectrum $M_2PN$ Deraining Spectrum

**Figure 6: The deraining quality comparison between these five best models.**

## 5 CONCLUSION

This paper proposes a lightweight multi-domain multi-attention progressive network ($M_2PN$) for a single image deraining. We analyze rain steaks' unique falling characteristics and special morphology, propose a frequency-channel attention (FcA) mechanism, and present a spatial-channel attention (ScA) mechanism to address deraining from spatial, frequency, and channel perspectives. The simple and recursive network structure and effective multi-domain multi-attention mechanism serve as the $M_2PN$ to achieve the overall best performance and make fast convergence during training in only 100 epochs. Furthermore, the $M_2PN$ structure, with a small network component quantity, shallow network channels, and few convolutional kernels, is a lightweight structure with only 168K parameters, which is much lower than the existing SOTA networks.

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
