# OpenReview forum: "A Lightweight Multi-domain Multi-attention Progressive Network for Single Image Deraining"
_acmmm.org/ACMMM/2024/Conference — MM2024 Poster_

### Official Review · Reviewer_uRKN · 2024-05-13

**Rating:** 4
**Confidence:** 4

**Summary:**

This paper proposes a lightweight multi-domain multi-attention progressive network (M2PN) to handle these challenges. For performance improvement, the M2PN backbone applies a simple progressive CNN-based structure consisting of the S same recursive M2PN modules. This recursive backbone with a skip connection mechanism allows for better gradient flow and helps to effectively capture low-to-high-level/scales spatial features in progressive structure to improve contextual information acquisition.
The method present the frequency-channel attention (FcA) mechanism and the spatial-channel attention (ScA) mechanism to fuse frequencychannel features and spatial features better to distinguish and remove rain steaks.

**Strengths:**

S1. The paper is well-organized and clearly written.
S1. The proposed network is lightweight and converges quickly.

**Limitations:**

L1. No comparison was made on real datasets.
L2. What is the reference basis for calculating the score in Table 2? Is it reasonable to consider M2PN as the best because PCNet does not have Rain1400's results?
L3. The evaluation index of WAAR method is SOTA in PSI/JNB, while the comparison between PSNR/SSIM index and it in this article is not significant, and PSI/JNB should be used instead.
L4. It should be compared with the latest algorithms, such as DRSformer(Learning A Sparse Transformer Network for Effective Image Deraining (CVPR 2023)).

**Suitability:**

3

---

### Official Review · Reviewer_gVur · 2024-05-22

**Rating:** 4
**Confidence:** 3

**Summary:**

This paper proposes a lightweight multi-domain multi-attention progressive network (M2PN) for a single image deraining. They analyze rain steaks’unique falling characteristics and special morphology, propose a frequency-channel attention (FcA) mechanism, and present a spatial-channel attention (ScA) mechanism to address deraining from spatial, frequency, and channel perspectives. The simple and recursive network structure and effective multi-domain multi-attention mechanism serve as the M2PN to achieve the overall best performance and make fast convergence during training in only 100 epochs. Furthermore, the M2PN structure, with a small network component quantity, shallow network channels, and few convolutional kernels, is a lightweight structure with only 168K parameters, which is much lower than the existing SOTA networks.

**Strengths:**

1)This paper theoretically explains the relationship between the spectral bandwidths and rain steaks’unique falling characteristics and special morphology, and practically proposes a FcA mechanism. This FcA can decompose and recombine the distribution of different frequency energies in the rainy image into K-bandwidths and allows for learning rain steak’s frequency information better to complement spatial information.
2)The ScA mechanism provides a fast closed-form energy solution to fuse the acquired frequency-channel features in a spatial-channel domain for better identification and removal of rain steaks.
3)The M2PN structure, with a small network component quantity, shallow network channels, and few convolutional kernels, is a lightweight structure with only 168K parameters, which is much lower than the existing SOTA networks.

**Limitations:**

1) Lack of comparison on real datasets;
2) It is not reasonable to compare the efficiency of the model only from the training epochs, rather than the inference time and calculation amount;
3) There is no comparison with enough other lightweight rain removal methods, and there is no comparison with the latest rain removal method in the comparison of rain removal effect.

**Suitability:**

2

---

### Official Review · Reviewer_rW4E · 2024-05-24

**Rating:** 2
**Confidence:** 4

**Summary:**

This article proposes a lightweight multi-domain multi-attention progressive network to address the issues of lightweight structure, low training, and testing efficiency. In addition, spectral analysis was conducted on the frequency energy distribution of rainbands, providing a theoretical understanding of the unique landing characteristics and special forms of rainbands and their relationship with spectral bandwidth. The experimental results indicate that the model can still achieve the best overall performance even with very few network parameters.

**Strengths:**

- The FcA mechanism was proposed to decompose and recombine the distribution of energy at different frequencies in rain images into k-bandwidth, which can better learn the frequency information of rain stripes to supplement spatial information.
- The ScA mechanism provides a fast closed-form energy solution that integrates the obtained frequency channel features into the spatial channel domain, thereby better identifying and removing rain streaks.
- The model achieved good performance with fewer parameters.

**Limitations:**

- The innovation of the paper is limited, and the global progressive backbone architecture is similar to PReNet and is not sufficient as an innovation point.
- The ablation experiment was insufficient, and the number of M2PN modules was not explored.
- $Conc (1×1)$in Figure 2 is unclear.
- We are curious about your evaluation criteria, why did you get the highest score when the performance was not optimal?
- We did not find Fig2-(a) mentioned in 3.1 in the article.

**Suitability:**

3

---

### Official Review · Reviewer_a3Ks · 2024-05-24

**Rating:** 3
**Confidence:** 4

**Summary:**

The paper proposes a lightweight multi-domain multi-attention progressive network (M2PN) for single image deraining. The M2PN uses a recursive CNN-based backbone with skip connections, where each module contains frequency-channel attention (FcA) and spatial-channel attention (ScA) mechanisms to extract complementary features in multiple domains. The FcA is based on spectral analysis of rain streaks using DCT, while the ScA provides a closed-form energy solution for spatial fusion. The network is designed to be lightweight, with only 168K parameters.

**Strengths:**

1. Novel integration of spectral analysis using DCT to characterize rain streaks in the frequency domain and inspire the FcA mechanism.
2. Multiple attention mechanisms (FcA, ScA) to extract complementary features in frequency, channel and spatial domains.
3. Lightweight design with only 168K parameters, achieved through ShiftAddNet convolutions and nearly parameter-free FcA/ScA.

**Limitations:**

1. The description of the FcA mechanism based on DCT analysis is interesting but lacks some technical details and ablation experiments to validate its individual contribution
2. Similarly, the closed-form solution in ScA is claimed to be inspired by an energy optimization principle, but the connection is not clearly explained.
3. While the overall structure is well-motivated, the exact architectural choices (e.g. number of modules S, blocks per group q, LSTM usage) could be further justified.
4. Missing some discussion on practical aspects like runtime/memory in comparison to heavier models.

**Suitability:**

2

---

### Meta-Review · Area_Chair_Q78N · 2024-06-30

**Recommendation:** Accept (Poster)
**Confidence:** 3

**Metareview:**

Upon reviewing the feedback provided by the reviewers, there are concerns regarding the absence of detailed technical explanations and a lack of comprehensive ablation studies and comparisons. Despite receiving “Borderline Accept” from three reviewers, I strongly believe this manuscript warrants further discussion to thoroughly address these concerns before reaching a final decision.